

# Elevation of troponin I in acute ischemic stroke

Yu-Chin Su[1], Kuo-Feng Huang[2,3], Fu-Yi Yang[1] and Shinn-Kuang Lin[1,3]

[1] Stroke Center and Department of Neurology, Taipei Tzu Chi Hospital, Buddhist Tzu Chi Medical Foundation, New Taipei City, Taiwan
[2] Department of Surgery, Division of Neurosurgery, Taipei Tzu Chi Hospital, Buddhist Tzu Chi Medical Foundation, New Taipei City, Taiwan
[3] School of Medicine, Tzu Chi University, Hualien, Taiwan

## ABSTRACT

**Background**. Cardiac morbidities account for 20% of deaths after ischemic stroke and is the second commonest cause of death in acute stroke population. Elevation of cardiac troponin has been regarded as a prognostic biomarker of poor outcome in patients with acute stroke.

**Methods**. This retrospective study enrolled 871 patients with acute ischemic stroke from August 2010 to March 2015. Data included vital signs, laboratory parameters collected in the emergency department, and clinical features during hospitalization. National Institutes of Health Stroke Scale (NIHSS), Barthel index, and modified Rankin Scale (mRS) were used to assess stroke severity and outcome.

**Results.** Elevated troponin I (TnI) > 0.01 µg/L was observed in 146 (16.8%) patients. Comparing to patients with normal TnI, patients with elevated TnI were older (median age 77.6 years vs. 73.8 years), had higher median heart rates (80 bpm vs. 78 bpm), higher median white blood cells (8.40 vs. 7.50 1,000/m$^3$) and creatinine levels (1.40 mg/dL vs. 1.10 mg/dL), lower median hemoglobin (13.0 g/dL vs. 13.7 g/dL) and hematocrit (39% vs. 40%) levels, higher median NIHSS scores on admission (11 vs. 4) and at discharge (8 vs. 3), higher median mRS scores (4 vs3) but lower Barthel index scores (20 vs. 75) at discharge ($p < 0.001$). Multivariate analysis revealed that age $\geq$ 76 years (OR 2.25, CI [1.59–3.18]), heart rate $\geq$ 82 bpm (OR 1.47, CI [1.05–2.05]), evidence of clinical deterioration (OR 9.45, CI [4.27–20.94]), NIHSS score $\geq$ 12 on admission (OR 19.52, CI [9.59–39.73]), and abnormal TnI (OR 1.98, CI [1.18–3.33]) were associated with poor outcome. Significant factors for in-hospital mortality included male gender (OR 3.69, CI [1.45–9.44]), evidence of clinical deterioration (OR 10.78, CI [4.59–25.33]), NIHSS score $\geq$ 12 on admission (OR 8.08, CI [3.04–21.48]), and elevated TnI level (OR 5.59, CI [2.36–13.27]). $C$-statistics revealed that abnormal TnI improved the predictive power of both poor outcome and in-hospital mortality. Addition of TnI > 0.01 ug/L or TnI > 0.1 ug/L to the model-fitting significantly improved $c$-statistics for in-hospital mortality from 0.887 to 0.926 ($p = 0.019$) and 0.927 ($p = 0.028$), respectively.

**Discussion.** Elevation of TnI during acute stroke is a strong independent predictor for both poor outcome and in-hospital mortality. Careful investigation of possible concomitant cardiac disorders is warranted for patients with abnormal troponin levels.

Corresponding author
Shinn-Kuang Lin,
jy0428@totalbb.net.tw

## INTRODUCTION

Heart disease and stroke are the second and third leading causes of death after cancer in Taiwan. Cerebrovascular and coronary artery diseases share many of the same risk factors. Cardiac mortality accounts for 20% of deaths and is the second commonest cause of death in the acute stroke population, second only to neurologic deaths as a direct result of the incident stroke (*Bounds et al., 1981*; *Prosser et al., 2007*). Prevalence of symptomatic and asymptomatic ischemic heart disease in acute stroke has been reported to be 20 to 30% and 40%, respectively (*Adams et al., 2003*). Cardiac troponins are important biomarkers of acute myocardial infarction and are routinely studied in the setting of ischemic heart disease. Abnormal levels of cardiac troponins have also been reported to be associated with poor clinical outcome in patients with acute cerebrovascular diseases, including ischemic stroke (*Di Angelantonio et al., 2005*; *Scheitz et al., 2012*; *Providência, Barra & Paiva, 2013*; *Faiz et al., 2014a*; *Faiz et al., 2014b*), intracerebral hemorrhage (*Hays & Diringer, 2006*), and spontaneous subarachnoid hemorrhage (*Deibert et al., 2003*).

Common risk factors for vascular diseases, such as hypertension, diabetes, heart disease, and hyperlipidemia, are well known comorbidities of stroke. Most previous studies emphasized the correlation of these comorbidities with stroke and clinical outcomes. However, the definition of each risk factor is usually not identical and the duration of these risk factors is not well described. The impact of a poorly controlled risk factor on the severity and outcome of stroke is not the same as that of a well-controlled one. Available laboratory parameters and clinical features as well as biomarkers during acute stroke provide valuable information when investigating the clinical outcomes after stroke. In this study, we investigated whether certain clinical features and laboratory parameters including troponin I (TnI) that are commonly measured on admission to the emergency department are predictive of outcome in patients with acute stroke.

## MATERIALS & METHODS

### Study population and data collection

Patients who were treated for stroke in the neurological ward from August 2010 to March 2015 were retrospectively selected from the stroke registry database. Inclusion criteria included a diagnosis of acute ischemic stroke that was confirmed by clinical presentation and proof of an ischemic lesion and/or absence of a corresponding intracranial lesion other than infarction by brain computed tomography or magnetic resonance study, and an available serum TnI study conducted in the emergency department within 48 h of symptom onset. TnI is considered as a routine study for patients with acute stroke by some but not all emergency physicians. Thus, measurement of TnI became a partial randomization study in the emergency department. Data integrated for analysis in this study included the sex and age of the patients, clinical data such as blood pressure and heart rate, and hematological parameters including the white blood cell count, hemoglobin, hematocrit, blood urea nitrogen, creatinine and TnI levels on arrival in the emergency department, and the severity of stroke evaluated on admission.

## Definitions

TnI was measured using a conventional VIDAS Troponin I Ultra assay (bioMerieux, Marcy L'Etoile, France) in the hospital's central laboratory. The analytical limit of detection and the 99th percentile upper reference limit was 0.01 µg/L. Abnormal elevation of TnI was defined as a TnI blood level >0.01 µg/L. Patients were stratified into two groups according to the TnI level, the normal group (≤ 0.01 µg/L) and the abnormal group (>0.01 µg/L). Patients with abnormal TnI levels were further stratified into two relatively equal-sized groups, the low-positive group (0.02–0.1 µg/L) and the high-positive group (>0.1 µg/L) (*Scheitz et al., 2014*). Stroke severity was assessed on admission according to the National Institutes of Health Stroke Scale (NIHSS). The etiology of stroke was classified according to the Trial of ORG 10172 in Acute Stroke Treatment (TOAST) criteria (*Adams et al., 1993*). Clinical deterioration was defined in patients who demonstrated an increase of two or more points in the NIHSS score during the acute stage of stroke (*Siegler et al., 2013*; *Umemura et al., 2014*). Outcomes were evaluated using the NIHSS, the Barthel index and the modified Rankin Scale (mRS) at discharge. An mRS score >2 was considered to indicate a poor outcome. All causes of death during hospitalization were registered as in-hospital mortality.

## Statistical analysis

Continuous variables are presented as median with interquartile range (IQR) or mean ± standard deviation. TnI values and mRS scores were analyzed as continuous and dichotomous variables. The chi-square test and Fisher's exact test were used for categorical comparisons of data. Group comparisons of continuous variables were performed using Mann–Whitney U and Kruskal–Wallis $H$ tests for independent samples. Significant predictors in the univariate analyses were transferred to dichotomous variables with the cut-off level according to the mean values of poor outcome, and were subsequently included in a multiple logistic-regression model to identify the most important factors associated with poor outcome, and in a stepwise logistic-regression model with in-hospital death. The predictive performance of the variables including TnI was compared using $c$-statistics. We compared basic models for poor outcome and in-hospital mortality including clinical variables to models that also included information on TnI levels. Comparisons of $c$-statistics were done according to the method of *DeLong, DeLong & Clarke-Pearson (1988)*. A $p$ value of less than 0.05 was considered to indicate statistical significance. All statistical analyses were performed with the statistical package SPSS (Version 17, SPSS Inc, Chicago, IL). The ROC curve comparisons were calculated using R software (version 2.15.3, pROC package). This study was approved by the Institutional Review Board of the Taipei Tzu Chi Hospital 04-XD40-107.

## RESULTS

During the study period, a total of 2,307 patients presented to our emergency department with acute ischemic stroke. Only 871 of those patients had valid data on TnI levels because during that period, measurement of TnI level was not routinely performed in the emergency department for patients with acute stroke. The average age of the 871

patients with valid TnI data was about 1.2 years older than that of all 2,307 patients (72.3 ± 13.6 vs. 71.1 ± 13.4 years, $p = 0.02$, Mann–Whitney test). Other baseline characteristics of the 871 patients with valid TnI data, including male-to-female ratio (1.13 vs. 1.21), percentage of patients with cardioembolism (15% vs. 13%), average blood pressure, heart rate, laboratory data, and NIHSS on admission (8.3 vs. 7.8), did not differ from those of the total 2,307 patients. The median age of the 871 patients enrolled in the study was 74.5 years (IQR 62.7–82.8) and 46.8% of them were women. The women were significantly older and had lower diastolic blood pressure, hemoglobin levels and hematocrit levels ($p < 0.001$) than the men. The women had higher NIHSS ($p = 0.024$) and mRS scores as well as lower Barthel index scores at discharge, and poorer outcomes ($p < 0.001$) than the men. The mortality rate was significantly higher in the men ($p = 0.017$). Elevated TnI levels were observed in 146 of the 871 patients (16.8%). Of these, 77 (8.8%) had high-positive levels and 69 (7.9%) had low-positive levels. Table 1 shows the comparison of clinical features, laboratory data, severity of stroke, and outcomes of patients with different levels of TnI. Abnormal TnI levels were more common in patients with stroke due to large artery atherosclerosis (54/232 = 23%), cardioembolism (38/131 = 29%), and undetermined etiology (5/17 = 29%) than in patients with stroke due to small vessel occlusion (48/482 = 10%) and other determined etiology (1/9 = 11%) according to the TOAST classification. Patients with abnormal TnI levels were significantly older ($p < 0.001$) than patients with normal TnI levels and had significantly higher heart rates ($p = 0.018$), white blood cell counts ($p = 0.025$), and creatinine levels ($p < 0.001$) and significantly lower hemoglobin ($p = 0.006$) and hematocrit levels ($p = 0.025$). In addition, patients with abnormal TnI levels had higher median NIHSS scores on admission (11, IQR 5–21) and at discharge (8, IQR 4–22) than patients with normal TnI levels (4, IQR 2–9 and 3, IQR 1–4, respectively) ($p < 0.001$). The median Barthel index was lower (20, IQR 0-75) and the mRS was higher (4, IQR 3–5) in patients with abnormal TnI than in patients with normal TnI (75, IQR 35–100 and 3, IQR 1–4, respectively) ($p < 0.001$). Poor outcomes were observed in 509 (58%) of the 871 patients and death occurred in 31 (3.6%) patients. Patients with abnormal TnI levels had longer hospital stays (16 days vs. 9 days), higher rates of clinical deterioration (18% vs. 9%, $p = 0.005$), poor outcome (79% vs. 54%, $p < 0.001$), and death (14% vs. 2%, $p < 0.001$) than patients with normal TnI levels. All the differences were more prominent in the high-positive group. There were mor poor outcomes (85%) and deaths (21%) in the high-positive group than in the low-positive group (74% and 6%, respectively).

Univariate analyses of continuous variables revealed that patients with poor outcomes were older, and had higher heart rates, TnI levels, white blood cell counts, creatinine levels, and NIHSS scores on admission and at discharge, and higher mRS scores at discharge but lower hemoglobin levels, hematocrit levels, and Barthel index scores than those with better outcomes (Table 2). Analysis of dichotomous variables revealed that female gender, cardioembolism, abnormal TnI levels, and evidence of clinical deterioration were associated with poor outcomes. In-hospital death was associated with high systolic blood pressure, high heart rate, high TnI level, high white blood cell count, and high NIHSS score on admission, and longer length of stay. Dichotomous analysis showed significant correlation

**Table 1  Correlation of clinical features and troponin I level in 871 patients with acute ischemic stroke.**

| Characteristics | Troponin I test[a] | | | Troponin I level (ug/L)[b] | | | |
|---|---|---|---|---|---|---|---|
| | Abnormal (>0.01 ug/L) (n = 146) | Normal (n = 725) | P value | >0.1 (n = 77) | 0.02–0.1 (n = 69) | ≥ 0.01 (n = 725) | P value |
| Median age (years) | 77.6 (66.2–85.6) | 73.8 (61.6–82.2) | <0.001 | 77.7 (67.2–84.9) | 77.5 (65.4–85.9) | 73.8 (61.6–82.2) | 0.003 |
| Systolic pressure (mmHg) | 160 (144–192) | 162 (144–184) | 0.972 | 157 (140–188) | 164 (145–195) | 162 (144–184) | 0.615 |
| Diastolic pressure (mmHg) | 87 (76–103) | 90 (79–101) | 0.294 | 87 (74–102) | 87 (76–106) | 90 (79–101) | 0.567 |
| Heart rate (bpm) | 80 (73–90) | 78 (67–89) | 0.015 | 84 (71–94) | 80 (74–88) | 78 (67–89) | 0.029 |
| White blood cells (1,000/mm$^3$) | 8.40 (6.48–7.03) | 7.50 (6.11–9.53) | 0.025 | 9.10 (7.11–11.01) | 7.44 (5.81–9.14) | 7.50 (6.11–9.53) | <0.001 |
| Hemoglobin (g/dL) | 13.0 (11.1–14.8) | 13.7 (12.3–14.9) | 0.006 | 12.9 (10.9–15.2) | 13.1 (11.2–14.6) | 13.7 (12.3–14.9) | 0.021 |
| Hematocrite (%) | 39.0 (34.0–43.2) | 40.0 (36.0–43.0) | 0.025 | 38.0 (33.8–44.0) | 39.0 (34.0–43.0) | 40.0 (36.0–43.0) | 0.074 |
| Glucose (mg/dL) | 144 (115–118) | 139 (112–185) | 0.612 | 156 (123–213) | 133 (113–158) | 139 (112–185) | 0.065 |
| Creatinine (mg/dL) | 1.40 (1.00–2.20) | 1.10 (0.90–1.30) | <0.001 | 1.40 (1.00–2.40) | 1.30 (1.0–2.0) | 1.10 (0.90–1.30) | <0.001 |
| NIHSS score (on admission) | 11 (5–21) | 4 (2–9) | <0.001 | 11 (6–23) | 10 (4–20) | 4.0 (2–9) | <0.001 |
| NIHSS score (at discharge) | 8 (4–22) | 3 (1–7) | <0.001 | 11 (4–32) | 6 (3–15) | 3 (1–7) | <0.001 |
| Barthel index score | 20 (0–75) | 75 (35–100) | <0.001 | 18 (0–66) | 30 (5–85) | 75 (35–100) | <0.001 |
| modified Rankin Scale | 4 (3–5) | 3 (1–4) | <0.001 | 5 (4–5) | 4 (1–5) | 3 (1–4) | <0.001 |
| Length of stay (days) | 16 (8–29) | 9 (5–24) | <0.001 | 19 (9–29) | 14 (6–28) | 9 (5–24) | <0.001 |
| Deterioration[c] | 26 (18%) | 68 (9%) | 0.005 | 16 (21%) | 10 (15%) | 68 (9%) | 0.006 |
| mRS > 2 [c] | 116 (79%) | 393 (54%) | <0.001 | 65 (85%) | 50 (74%) | 393 (54%) | <0.001 |
| Death [c] | 20 (14%) | 11 (2%) | <0.001 | 16 (21%) | 4 (6%) | 11 (2%) | <0.001 |

**Notes.**

[a] Mann–Whitney U test.
[b] Kruskal-Wallis test.
[c] Chi-square test.
NIHSS, National Institute of Health Stroke Scale; mRS, modified Rankin Scale.
Data are expressed as median (IQR) or n (%).

**Table 2  Correlation of clinical features and outcomes in 871 patients with acute ischemic stroke.**

| Characteristics | Poor outcome (mRS >2)[a] | | | | P value | Death[a] | | P value |
|---|---|---|---|---|---|---|---|---|
| | Mean | | Median | | | Median | | |
| | Y (n = 509) | N (n = 362) | Y (n = 509) | N (n = 362) | | Y (n = 31) | N (n = 840) | |
| Age (years) | 75.7 ± 12.5 | 67.4 ± 13.5 | 78.0 (67.3–85.1) | 66.9 (57.5–78.3) | <0.001 | 78.0 (69.3–84.3) | 74.5 (62.5–82.7) | 0.180 |
| Systolic pressure (mmHg) | 165 ± 32 | 164 ± 29 | 162 (144–186) | 163 (144–184) | 0.904 | 182 (150–200) | 162 (144–184) | 0.034 |
| Diastolic pressure (mmHg) | 89 ± 18 | 92 ± 18 | 88 (77–102) | 91 (80–101) | 0.114 | 86 (72–106) | 89 (79–101) | 0.972 |
| Heart rate (bpm) | 82 ± 17 | 77 ± 16 | 80 (69–92) | 76 (66–85) | <0.001 | 84 (78–97) | 78 (67–89) | 0.019 |
| Troponin I (ug/L) | 0.119 ± 0.656 | 0.022 ± 0.101 | 0.01 (0.01–0.01) | 0.01 (0.01–0.01) | <0.001 | 0.09 (0.01–0.65) | 0.01 (0.01–0.01) | <0.001 |
| White blood cells (1,000/mm$^3$) | 8.32 ± 3.03 | 7.84 ± 2.65 | 7.81 (6.20–9.98) | 7.40 (6.19–9.26) | 0.039 | 9.00 (6.71–11.80) | 7.60 (6.20–9.61) | 0.022 |
| Hemoglobin (g/dL) | 13.1 ± 2.2 | 13.9 ± 2.0 | 13.3 (11.7–14.5) | 14.0 (12.7–15.2) | <0.001 | 13.5 (11.3–14.9) | 13.6 (12.2–14.9) | 0.698 |
| Hematocrite (%) | 38.7 ± 5.9 | 40.7 ± 5.1 | 39 (35–43) | 41 (38–44) | <0.001 | 39 (35–43) | 40 (36–43) | 0.631 |
| Glucose (mg/dL) | 169 ± 88 | 164 ± 84 | 144 (115–195) | 134 (109–187) | 0.054 | 162 (136–205) | 139 (113–194) | 0.130 |
| Creatinine (mg/dL) | 1.52 ± 1.47 | 1.30 ± 1.02 | 1.2 (0.9–1.5) | 1.1 (0.9–1.3) | 0.032 | 1.5 (1.1–2.0) | 1.1 (0.9–1.4) | <0.001 |
| NIHSS score (on admission) | 12.2 ± 9.0 | 2.9 ± 3.0 | 9 (5–18) | 2 (1–4) | <0.001 | 25 (20–32) | 5 (2–10) | <0.001 |
| NIHSS score (at discharge) | 11.7 ± 11.4 | 1.4 ± 1.6 | 7 (4–16) | 1 (0–2) | <0.001 | – | – | |
| Barthel index score | 34.5 ± 28.9 | 95.7 ± 9.3 | 35 (5–60) | 100 (95–100) | <0.001 | – | – | |
| Modified Rankin Scale | 4.2 ± 0.9 | 0.9 ± 0.6 | 4 (4–5) | 1 (1–1) | <0.001 | – | – | |
| Length of stay (days) | 23.2 ± 16.8 | 6.8 ± 5.2 | 20 (10–30) | 5 (4–8) | <0.001 | 9 (3–24) | 11 (5–25) | 0.213 |
| Male gender[b] | | | 243 (48%) | 220 (61%) | <0.001 | 23 (74%) | 440 (52%) | 0.017 |
| Cardioembolism[b] | | | 95 (19%) | 36 (10%) | <0.001 | 7 (23%) | 124 (15%) | 0.301 |
| Abnormal troponin I[b] | | | 116 (23%) | 30 (8%) | <0.001 | 20 (65%) | 126 (15%) | <0.001 |
| Deterioration[b] | | | 86 (17%) | 8 (2%) | <0.001 | 18 (58%) | 76 (9%) | <0.001 |

**Notes.**

[a] Mann–Whitney *U* test.

[b] Chi-square test.

mRS, modified Rankin Scale; IQR, interquartile range; NIHSS, National Institute of Health Stroke Scale.

Data are expressed as mean ± sd, median (IQR) or *n* (%).

**Table 3** Regression model of factors influencing outcomes and mortality in 871 patients with acute ischemic stroke.

| Characteristics | Poor outcome (mRS > 2)[a] | | Death[b] | |
|---|---|---|---|---|
| | OR (95% CI) | P value | OR (95% CI) | P value |
| Male gender | 0.75 (0.53–1.06) | 0.108 | 3.69 (1.45–9.44) | 0.006 |
| Age ≥ 76 years | 2.25 (1.59–3.18) | <0.001 | | |
| Heart rate ≥ 82 bpm | 1.47 (1.05–2.05) | 0.026 | | |
| White blood cells ≥8320 uL | 1.21 (0.86–1.70) | 0.264 | | |
| Hemoglobin ≤ 13.1 g/dL | 1.44 (0.78–2.65) | 0.245 | | |
| Hematocrite ≤ 38.7% | 1.22 (0.68–2.23) | 0.508 | | |
| Creatinine ≥ 1.52 mg/dL | 1.05 (0.67–1.64) | 0.848 | | |
| Cardioembolism | 0.85 (0.50–1.46) | 0.573 | | |
| Deterioration | 9.45 (4.27–20.94) | <0.001 | 10.78 (4.59–25.33) | <0.001 |
| NIHSS score (admission) ≥12 | 19.52 (9.59–39.73) | <0.001 | 8.08 (3.04–21.48) | <0.001 |
| Troponin I > 0.01 ug/L | 1.98 (1.18–3.33) | 0.010 | 5.59 (2.36–13.27) | <0.001 |

Notes.

NIHSS, National Institute of Health Stroke Scale; mRS, modified Rankin Scale; OR, odds ratio; CI, confidence interval.

[a] Multiple logistic regression.

[b] Stepwise backward regression.

of male gender, abnormal TnI levels, and evidence of clinical deterioration with in-hospital death. Age was not associated with in-hospital death.

Table 3 showed the regression analysis of the significant dichotomous variables with the cut-off levels according to the mean values of poor outcome in Table 2. Multivariate logistic regression analysis revealed that age ≥ 76 years ($p < 0.001$), heart rate ≥ 83 bpm ($p = 0.001$), evidence of clinical deterioration ($p < 0.001$), NIHSS score ≥14 on admission ($p < 0.001$), and abnormal TnI level (odds ratio [OR]: 1.98; 95% confidence interval CI [1.18–3.33]; $p = 0.01$) were significant predictors of poor outcome. A stepwise backward regression analysis showed male gender ($p = 0.006$), evidence of clinical deterioration ($p < 0.001$), NIHSS score ≥ 12 on admission ($p < 0.001$), and abnormal TnI level (OR: 5.59; 95% CI [2.36–13.27]; $p < 0.001$) were significant predictors of in-hospital mortality.

C-statistics of regression models for detection of poor outcome and death for each factor are shown in Table 4. C-statistics for detection of poor outcome was 0.691 for NIHSS score ≥12 on admission. The addition of age ≥76 years, evidence of clinical deterioration, and TnI ≥ 0.01 ug/L to the regression model resulted in significant improvement of the c-statistics to 0.787 ($p < 0.05$). The addition of TnI > 0.01 ug/L or TnI > 0.1 ug/L to a model-fitting including significant factors in logistic regression (NIHSS score on admission ≥12, age ≥76 years, evidence of clinical deterioration, heart rate ≥ 82 bpm) did not improve the predictive value foro poor outcome. Similar results were observed for detection of death. C-statistics for detection of death was 0.790 for an NIHSS score ≥12 on admission. The addition of TnI ≥ 0.01 ug/L and evidence of clinical deterioration to the regression model resulted in significant improvement of the c-statistics to 0.912 ($p < 0.05$). The addition of TnI > 0.01 ug/L or TnI >0.1 ug/L to a model-fitting including NIHSS ≥12 on admission, evidence of deterioration and male gender significantly improved the predictive value for death from 0.887 to 0.926 and 0.927, respectively ($p < 0.05$).

**Table 4   *C*-statistics for prediction of poor outcome and in-hospital mortality.**

| Poor outcome (mRS > 2)[a] | | | Death[b] | | |
|---|---|---|---|---|---|
| Characteristics | *C*-statistics (95% CI) | *P*[c] | Characteristics | *C*-statistics (95% CI) | *P*[c] |
| NIHSS score (admission) $\geq$ 12 | 0.691 (0.657–0.725) | | NIHSS score (admission) $\geq$ 12 | 0.790 (0.707-0.872) | |
| Includes age $\geq$ 76 years | 0.748 (0.717–0.780) | <0.001 | Includes troponin I > 0.01ug/L | 0.860 (0.799–0.921) | 0.001 |
| Further includes deterioration | 0.778 (0.748–0.808) | <0.001 | Further includes deterioration | 0.912 (0.859–0.965) | 0.018 |
| Further includes troponin I > 0.01 ug/L | 0.787 (0.758–0.817) | 0.006 | Further includes male gender | 0.926 (0.884–0.969) | 0.134 |
| Further includes heart rate $\geq$ 82 bpm | 0.796 (0.767–0.825) | 0.057 | | | |
| Model 1 | 0.790 (0.761–0.819) | *P*[d] | Model 2 | 0.887 (0.829–0.946) | *P*[d] |
| +Troponin I > 0.01ug/L | 0.796 (0.767–0.825) | 0.155 | +Troponin I > 0.01ug/L | 0.926 (0.884–0.969) | 0.019 |
| +Troponin I > 0.1ug/L | 0.798 (0.769–0.826) | 0.106 | +Troponin I > 0.1ug/L | 0.927 (0.886–0.968) | 0.028 |

**Notes.**

mRS, modified Rankin Scale; NIHSS, National Institute of Health Stroke Scale; CI, confidence interval.

[a] Multiple logistic regression.

[b] Stepwise backward regression.

[c] Compared with previous one.

[d] Compared with Model 1 or 2.

Model 1 includes NIHSS score (admission) $\geq$ 12, age $\geq$ 76 years, deterioration, heart rate $\geq$ 82 bpm Model 2 includes NIHSS score (admission) $\geq$ 12, deterioration, male gender.

## DISCUSSION

TnI is a highly sensitive and specific marker of acute myocardial infarction. Elevated TnI is characteristic of a number of cardiac diseases as well such as heart failure, pericarditis, myocarditis, atrial fibrillation and tachycardia (*Tanindi & Cemri, 2011*). Elevated TnI has also been found in patients with chronic renal failure, sepsis, critical illness, pulmonary embolism, chronic obstructive pulmonary disease, and stroke (*Tanindi & Cemri, 2011*; *Mannu, 2014*). Elevated levels of cardiac troponin have been reported in 10–34% of patients with acute stroke. *Kerr et al. (2009)* conducted a systematic review of studies measuring troponin within 7 days of symptom onset in acute stroke patients and found that more than 18% of patients had a high troponin level. Some studies reported that elevated troponin levels were more common in patients with stroke due to cardioembolism who also had evidence of atrial fibrillation, ischemic heart or heart failure (*Etgen et al., 2005*; *Faiz et al., 2014a*; *Faiz et al., 2014b*). Abnormal TnI levels were observed in 16.8% patients in our study. We found that patients with abnormal TnI were more likely to have large artery atherosclerosis, cardioembolism and undetermined etiology. Patients who had risks from both atrial fibrillation and stenotic cerebral arteries were grouped into undetermined etiology with conflicting data when categorizing the subtype of stroke. This could explain why there was a similarly higher percentage of elevated TnI levels in patients with undetermined etiology. Patients with elevated TnI levels were older and had higher heart rates and creatinine levels but lower hemoglobin levels and hematocrits than patients with normal TnI levels. Patients with elevated TnI presented with more severe initial stroke severity and showed a greater degree of clinical deterioration during hospitalization. Worse outcomes and higher in-hospital mortality were observed in patients with abnormal TnI as well. All of the above differences were most prominent in patients with high-positive TnI levels. These findings are similar to those reported by Di Angelantonio et al., who found a dose–response relationship between the three TnI groups (normal, low-positive, and high-positive) and clinical features (*Di Angelantonio et al., 2005*).

Mechanisms for elevated TnI during acute ischemic stroke may be separated into 2 major groups, (1) ischemic myocardial injury (ie, because of coronary ischemia) and (2) nonischemic (noncoronary) myocardial injury (*Scheitz et al., 2015a*). Coronary plaque rupture or mismatch between oxygen demand and supply (such as in tachyarrhythmia, hypertensive crisis, or respiratory failure) may cause ischemic myocardial injury. Nonischemic myocardial injury comprises neurogenic heart syndrome and noneurogenic conditions (severe infection or sepsis, heart or renal failure, pulmonary embolism). In neurogenic heart syndrome, acute stroke-related increased sympathetic activity with excessive catecholamine release results in coagulative myocytolysis (also known as contraction band necrosis or myofibrillar degeneration) or cardiomyopathy. Myocytolysis surrounding patches of subendocardial hemorrhage or swollen myocytes surrounding epicardiac nerves during early acute stroke has been suggested to be the cause of cardiac injury (*Oppenheimer & Hachinski, 1992*). *Barber et al. (2007)* found that raised TnI was associated with elevation of circulating epinephrine in patients with acute ischemic stroke. Involvement of the parietal lobe or insular cortex has also been associated with elevated cardiac troponin levels due to the

imbalance of sympathetic and parasympathetic autonomic control (*Ay et al., 2006*; *Rincon et al., 2008*; *Scheitz et al., 2015b*). Not all patients in our study underwent brain magnetic resonance imaging to indentify the precise location of the stroke; therefore, we were not able to analyze the involvement of the insular or parietal cortex. Cardiac cell damage with elevated troponins in acute stroke may be enhanced by the stress-related inflammatory response as well as the cytokine response pathways (*Christensen et al., 2004*). The etiologies of elevated troponin levels other than acute coronary syndrome in renal failure include subclinical myocardial damage (micro-infarctions) and decreased renal troponin excretion (*Freda et al., 2002*; *Jensen et al., 2007*; *Faiz et al., 2014a*; *Faiz et al., 2014b*). Serum troponin T is increased more frequently than TnI in patients with renal failure, and TnI has been reported to be a more sensitive and specific biomarker of cardiac damage than Troponin T in patients with end-stage renal failure (*Freda et al., 2002*; *Mannu, 2014*).

There is no doubt that advanced age, higher NIHSS score on admission, and evidence of clinical deterioration during hospitalization are associated with a worse outcome and higher death rate at discharge. The average age of patients with abnormal TnI, patients with poor outcome, and patients who died in the hospital in this study was approximately 76 years. *Faiz et al. (2014a)* and *Faiz et al. (2014b)* also reported that age $\geq$ 76 years was independently associated with elevated troponin levels in patients with acute ischemic stroke. A high heart rate was associated with worse outcomes, in particular death, in a long-term follow-up study of patients with vascular diseases (*Erdur et al., 2014*). In our study, a higher heart rate was observed in patients with abnormal TnI (83 bpm), in those with poor outcome (82 bpm), and in those who died before discharge (87 bpm) than those without these factors. Multivariate analysis revealed that heart rate $\geq$82 bpm was also an independent risk factor for poor outcome. *Erdur et al. (2014)* reported that heart rate $\geq$83 bpm on admission was independently associated with in-hospital mortality in acute ischemic stroke patients, suggesting early negative effects of autonomic imbalance.

With the exception of the NIHSS score on admission and subsequent deterioration during hospitalization, only elevated TnI was a strong independent predictor of both poor outcome and death. Abnormal TnI had an OR of 1.98 for poor outcome and an OR of 5.59 for in-hospital mortality. A meta-analysis of 2901 patients from 15 studies with different definitions and sampling times for troponin by *Kerr et al. (2009)* revealed that elevated troponin is associated with poor outcome; however, they did not fully establish whether elevated troponin is an independent prognostic factor. Recent studies with multivariate models including age and some measures of stoke severity have concluded that a positive level of troponin is associated with an overall increased risk of both death and disability (*Jensen et al., 2007*; *Faiz et al., 2014a*; *Faiz et al., 2014b*). In the present study, *c*-statistics revealed that abnormal TnI improved the predictive power of both poor outcome and in-hospital mortality. Although the predictive performance of abnormal TnI for poor outcome was relative low in a categorized model with strong predictors, such as NIHSS score $\geq$ 12 on admission and evidence of deterioration, both low-positive and high-positive TnI significantly increased the discriminative power of the model for in-hospital death. The American Stroke Association recommends the routine checking of markers of cardiac ischemia during acute stroke (*Adams et al., 2007*). Whether troponin should be routinely

checked is still under deliberation. Nevertheless, recognition and careful investigation of possible concomitant cardiac disorders in patients with acute ischemic stroke is warranted for patients with elevated troponin levels.

The newly developed high-sensitivity assay of troponin allows for precise detection of troponin even at concentrations 10-fold lower than conventional assays (*Wu & Jaffe, 2008*). A high-sensitivity troponin test improves the diagnosis of patients with acute myocardial infarct. However, reduction of specificity comes with improvement in sensitivity. In Scheitz's series, troponin T elevation above the 99th percentile was detected with a high-sensitivity assay in more than 50% of patients with acute ischemic stroke, and even moderately elevated troponin T was associated with an unfavorable outcome (*Scheitz et al., 2014*). The presence of high positive or dynamic change troponin levels might indicate ischemic myocardial injury. Stroke patients with dynamic changes in troponin levels (>50%) within 24 h showed a higher risk for in-hospital mortality than patients with increased troponin levels who were stable over time (*Scheitz et al., 2014*). Serial measurements should be performed to establish whether troponin is acutely or chronically elevated. For patients with non-acute elevation of troponin levels, out-patients evaluation for structural or coronary heart disease is recommended. For patients with high positive or a dynamic pattern of elevated troponin levels, prompt measures for prevention of cardiovascular disease should be intensified or reevaluated. Noninvasive echocardiography, cardiac magnetic resonance imaging or computed tomography may help to identify possible unstable coronary disease, heart failure, or cardiomyopathy. Invasive coronary angiography may be indicated for patients with acute myocardial infarction (*Scheitz et al., 2015a*).

This study has a number of limitations. First, this study was retrospective in nature. There was selection bias exits because of only a small group of patents received TnI measurement. Second, TnI was checked only once in each patient in the emergency room without an exact time period of onset-of-symptoms to troponin measurement. Dynamic change measurement of TnI might provide more prognostic relevance in acute ischemic stroke. Third, the low number of patients with the outcome "death" might limit meaningfulness although stepwise logistic regression analysis was used. Finally, we did not perform a follow-up study after discharge. Notwithstanding these limitations, our data extend the current understanding of the implications of troponin positivity in acute ischemic stroke.

## CONCLUSIONS

Troponin I provides better information than age and other laboratory parameters in the prediction of the outcome of stroke, even after adjustment for the strong impact factors of stroke severity and presence of clinical deterioration. Elevation of TnI during acute stroke is a strong independent predictor of both poor outcome and in-hospital mortality. Both neurologists and cardiologists need to pay more attention to possible concomitant cardiac disorders in patients with abnormal troponin levels during acute stroke.

### Funding

The study was supported by a grant from the Taipei Tzu Chi Hospital, Buddhist Tzu Chi Medical Foundation (TCRD-TPE-103-RT-16). The funders had no role in study design, data collection and analysis, decision to publish, or preparation of the manuscript.

### Grant Disclosures

The following grant information was disclosed by the authors:
Taipei Tzu Chi Hospital, Buddhist Tzu Chi Medical Foundation: TCRD-TPE-103-RT-16.

### Competing Interests

The authors declare there are no competing interests.

### Author Contributions

- Yu-Chin Su performed the experiments, wrote the paper, prepared figures and/or tables, reviewed drafts of the paper.
- Kuo-Feng Huang performed the experiments, contributed reagents/materials/analysis tools, reviewed drafts of the paper.
- Fu-Yi Yang performed the experiments, reviewed drafts of the paper.
- Shinn-Kuang Lin conceived and designed the experiments, performed the experiments, analyzed the data, contributed reagents/materials/analysis tools, wrote the paper, prepared figures and/or tables, reviewed drafts of the paper.

### Human Ethics

The following information was supplied relating to ethical approvals (i.e., approving body and any reference numbers):

This study was approved by the Institutional Review Board of the Taipei Tzu Chi Hospital 04-XD40-107.

### Data Availability

The raw data was supplied as Supplemental Information 1.

### Supplemental Information

Supplemental information for this article can be found online at http://dx.doi.org/10.7717/peerj.1866#supplemental-information.

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
