# Peer review of "Elevation of troponin I in acute ischemic stroke"

_PeerJ, doi:10.7717/peerj.1866_

## Round 0.1 · original submission · Major Revisions

In addition to responding to the reviewer comments, please clarify novelty of your study and respond to the main points regarding potential selection bias, the dichotomization by Tn levels and the criteria for neurological deterioration you used. Please also cite the online review in Stroke.

·

Basic reporting

Elevation of Troponin I in Acute Ischemic Stroke

SU et al., present an observational, retrospective study (of a selected hospital population) that addresses the association between elevated troponin I and prognosis in ischemic stroke patients. The authors find that elevated troponin I is independently associated with outcome (in terms of deaths and unfavorable outcome). The manuscript is well written. Results are mainly confirmatory. Several previous studies have presented data on the topic, a recent review is online in STROKE.

The following points may help to further improve the paper/need to be addressed before publication.

Selection bias: 2.307 patients with acute ischemic stroke were admitted to the hospital, but only 871 received TropI measurements. A possible explanation would be selection by indication. This needs to be addressed as a limitation. Ideally, the selection bias would be described. (“Patients with Trop I measurements were older, more often, female, more often had history of CHD……”)

Point out differences in sensitivity between “conventional” and “high sensitive” Troponin tests. Type of test may influence frequency of elevated values. Name the assay used to testing Troponin. In cohorts of patients with ischemic stroke, up to 35% have measurable cTn if conventional assays are used. Up to 60% have elevated cTn levels when cTn is measured with highly sensitive assays.

Abstract
Rather than giving p-values for only some of the variables significantly different in univariate testing, (no values for NIHSS, mRS given in the abstract), omit p-values in the abstract and give OR [CI] of multivariate testing only. (Easier to read).

Introduction
The problem, that risk factor definition may follow differing definitions in different studies is pointed out in the introduction but of no further relevance to the paper itself. This aspect is not dealt with. Does this information add anything to the message of the paper?

Methods:
Please clarify Does “[stroke] confirmed by brain computed tomography” mean prove of ischemic lesion or absence of intracerebral bleeding / further differential diagnosis?

Is data available on the time period “onset-of-symptoms to troponin measurement”? Time of measurement may influence likelihood of detection of elevated conecentrations. (Possible limitation).

Definition:
Plese clarify definition of “low positive group”: “>0,01µg/L” (including very high elevations) or >0,01µg/L but <0,1µg/L (excluding very high elevations).

What was the rational to define “deterioration” as more than two points on the NIHSS? Any reference for that?

Outcome:
A mRS >3 is unusual as outcome definition. Rather use mRS >=2 (>1) or give reason for this cut-off.

Is there data on hospital lengths-of-stay (which influences likelihood to detect death)?

Written informed consent:
How many patients (or relatives) did not give informed consent? Does a screen log exist? Important in terms of selection bias.

Results:
Paragraph starting line 168:
Give motivation for dichotomizing age at 76 years, heart rate at 83 bpm……
Were these cut-offs chosen in advance?

Discussion:
First paragraph: Gender is not the main topic of the paper. Omit the first paragraph on gender disparities. Start with the main finding of the paper (with regard to the aim).

Line 216: “levels” is plural, correct verb form = have.

Line 236: Please refer to Scheitz et al. Stroke 2015a Application and interpretation of hs-TnT in acute stroke. The review discusses mechanisms of trop elevation in stroke in detail.

Line 245: Scheitz et al. 2015b reports on insular cortex lesion and troponin (“Insular cortex lesions, cardiac troponin, and detection of previously unknown atrial fibrillation in acute ischemic stroke: insights from the troponin elevation in acute ischemic stroke study.“)
Line 284: Application and interpretation of high-sensitivity cardiac troponin assays in patients with acute ischemic stroke. Stroke. 2015;46:1132-4 makes a detailed suggestion how to handle the problem.
Line 286: Dynamics of Troponin may give more information on origin and prognosis than a single measurement alone. Please acknowledge existing data: Prognostic relevance of cardiac troponin T levels and their dynamic changes measured with a high-sensitivity assay in acute ischaemic stroke: analyses from the TRELAS cohort. Int J Cardiol 2014

Line 287: Excluding patients with concomitant MI or impaired renal function does not necessarily improve the analysis – adjustment is elementary.

Limitations:
Please name – and if possible describe in the results section - selection bias.

Please point out, that low number of the outcome "deaths" limits meaningfulness of multivariate regression analysis.

Minor comment:
Results, line 36: replace “patients of different genders” to “according to gender”.

Experimental design

The design is sound.
However, number of deaths (N=31) limits multivariate regression Analysis (for deaths) because it implies a risk for "overfitting". In general, you should have N=10 outcomes per 1 variable in the model. [In this case, you should limit variables to three (N=31 outcomes), but use eleven].
You should think about using a stepweise backwards Approach and should add "small number to deaths" as limitation.

Validity of the findings

See above.

Additional comments

Elevation of Troponin I in Acute Ischemic Stroke

SU et al., present an observational, retrospective study (of a selected hospital population) that addresses the association between elevated troponin I and prognosis in ischemic stroke patients. The authors find that elevated troponin I is independently associated with outcome (in terms of deaths and unfavorable outcome). The manuscript is well written. Results are mainly confirmatory. Several previous studies have presented data on the topic, a recent review is online in STROKE.

The following points may help to further improve the paper/need to be addressed before publication.

Selection bias: 2.307 patients with acute ischemic stroke were admitted to the hospital, but only 871 received TropI measurements. A possible explanation would be selection by indication. This needs to be addressed as a limitation. Ideally, the selection bias would be described. (“Patients with Trop I measurements were older, more often, female, more often had history of CHD……”)

Point out differences in sensitivity between “conventional” and “high sensitive” Troponin tests. Type of test may influence frequency of elevated values. Name the assay used to testing Troponin. In cohorts of patients with ischemic stroke, up to 35% have measurable cTn if conventional assays are used. Up to 60% have elevated cTn levels when cTn is measured with highly sensitive assays.

Abstract
Rather than giving p-values for only some of the variables significantly different in univariate testing, (no values for NIHSS, mRS given in the abstract), omit p-values in the abstract and give OR [CI] of multivariate testing only. (Easier to read).

Introduction
The problem, that risk factor definition may follow differing definitions in different studies is pointed out in the introduction but of no further relevance to the paper itself. This aspect is not dealt with. Does this information add anything to the message of the paper?

Methods:
Please clarify Does “[stroke] confirmed by brain computed tomography” mean prove of ischemic lesion or absence of intracerebral bleeding / further differential diagnosis?

Is data available on the time period “onset-of-symptoms to troponin measurement”? Time of measurement may influence likelihood of detection of elevated conecentrations. (Possible limitation).

Definition:
Plese clarify definition of “low positive group”: “>0,01µg/L” (including very high elevations) or >0,01µg/L but <0,1µg/L (excluding very high elevations).

What was the rational to define “deterioration” as more than two points on the NIHSS? Any reference for that?

Outcome:
A mRS >3 is unusual as outcome definition. Rather use mRS >=2 (>1) or give reason for this cut-off.

Is there data on hospital lengths-of-stay (which influences likelihood to detect death)?

Written informed consent:
How many patients (or relatives) did not give informed consent? Does a screen log exist? Important in terms of selection bias.

Results:
Paragraph starting line 168:
Give motivation for dichotomizing age at 76 years, heart rate at 83 bpm……
Were these cut-offs chosen in advance?

Discussion:
First paragraph: Gender is not the main topic of the paper. Omit the first paragraph on gender disparities. Start with the main finding of the paper (with regard to the aim).

Line 216: “levels” is plural, correct verb form = have.

Line 236: Please refer to Scheitz et al. Stroke 2015a Application and interpretation of hs-TnT in acute stroke. The review discusses mechanisms of trop elevation in stroke in detail.

Line 245: Scheitz et al. 2015b reports on insular cortex lesion and troponin (“Insular cortex lesions, cardiac troponin, and detection of previously unknown atrial fibrillation in acute ischemic stroke: insights from the troponin elevation in acute ischemic stroke study.“)
Line 284: Application and interpretation of high-sensitivity cardiac troponin assays in patients with acute ischemic stroke. Stroke. 2015;46:1132-4 makes a detailed suggestion how to handle the problem.
Line 286: Dynamics of Troponin may give more information on origin and prognosis than a single measurement alone. Please acknowledge existing data: Prognostic relevance of cardiac troponin T levels and their dynamic changes measured with a high-sensitivity assay in acute ischaemic stroke: analyses from the TRELAS cohort. Int J Cardiol 2014

Line 287: Excluding patients with concomitant MI or impaired renal function does not necessarily improve the analysis – adjustment is elementary.

Limitations:
Please name – and if possible describe in the results section - selection bias.

Minor:
Results, line 36: replace “patients of different genders” to “according to gender”.

·

Basic reporting

No Comments

Experimental design

MAJOR COMMENTS:
- My main concern in this study is that, from 2,307 patients included during the study period, Tn is measured in just 871 patients. Authors should clarify why Tn was measured in these cases, as they may represent a selected, high risk population (perhaps Tn is just measured if a cardiac comorbidity is observed or suspected). In this sense, a comparison of the baseline characteristics and comorbidities between patients with and without Tn would be useful.
MINOR POINTS
- Please give a rationale for further dichotomization of elevated Tn levels into high and low).
- Please justify and give a reference about why clinical deterioration was defined as an increase on the NIHSS>2 points, as usually >4 points are required.

Validity of the findings

MAJOR COMMENTS:
- An assessment of the additional predictive value of Tn in the prediction of mortality and disability after stroke over clinical information would be of great interest. Please consider comparison of C-statistics of performance of the integrated discrimination index.
MINOR POINTS
- Were cardiac comorbidities included in the logistic regression models? If no, please include.
- Were continuous variables tested for normality? If not normal, please report as median (IQR) and use appropriate statistical methods.
- Please give mean/median in-hospital stay, as all the endpoints are assessed at discharge.
- Tn was obtained within the first 48 hours, a huge range of time. If possible, could the authors adjust the results in any way considering time from symptom onset to blood withdrawal?

Additional comments

MAJOR COMMENTS:
- The study lacks novelty. The association of Tn with in mortality and outcome after stroke has been repeatedly reported. What are the novel findings of this study?
MINOR COMMENTS
- The discussion should be more focused on a translational point of view. What do the authors think that a clinician should do with an elevated Troponin?
- Authors said in the abstract and introduction that cardiac morbidities account for a 20% of deaths after ischemic stroke, but these data are based on a study from 1981. Please update.
- Table 1 and first paragraph of the discussion, although could be of interest, are not related with the endpoints of the study: Please consider removal or inclusion as supplementary-online data.
- Table 3: the term 'deteriotaion' is misspelled.
- Table 4: please change OD x OR as odds ratio abbreviation

---

## Round 0.2 · Minor Revisions

Please address Reviewer 2's request for further minor revisions.

·

Basic reporting

Well written manuscript.

Experimental design

No additional comments.

Validity of the findings

No additional comments.

Additional comments

The authors have addressed may comments sufficiently.

·

Basic reporting

I have no comments

Experimental design

1. Author's addess the limitation regarding Tn measure in just 871 out of 2,307. However, an additional effort should be performed to compare baseline characteristics of patients with and without Tn measures. If data from patients without TN measures are not available, this should be mentioned as a limitation.
2. The authors provide some references justifying the choice of a cut-off point of 2 for the NIHSS as a definition of clinical deterioration in the comments to reviewers. I would suggest to include some of these references in the manuscript, as clinicians from Europe or America are not used to this definition.
3. Please provide a reference for the choice of the > 0.1μg/L cut-off. Was it based on the manufacturer’s results or in previous literature, or arbitrarily selected?

Validity of the findings

I have no additional comments.

Additional comments

Although results are still mainly confirmatory, the manuscript has improved with the review process. The assessment of the additional predictive value of Tn by the C-statistics comparison by theLong's method is a major study finding and adds novelty. Please consider the minor points abovementioned and carefully check english.

---

## Round 0.3 · accepted · Accept

Thank you for your further revisions. The manuscript is now acceptable for publication.